# Blood Donor RBC Genotyping at the National Advisory Unit on Immunohematology at Oslo University Hospital

**DOI:** 10.3390/biomedicines13122907

**Published:** 2025-11-27

**Authors:** Inger Margit Alm, Hoyi Wong, Monica Stensrud, Geir Tomter, Mette S. Bævre, Çiğdem Akalın Akkök

**Affiliations:** Department of Immunology and Transfusion Medicine, Oslo University Hospital, Ullevaal, N-0407 Oslo, Norway; ingermargitalm@outlook.com (I.M.A.); hoywon@ous-hf.no (H.W.); uxsmom@ous-hf.no (M.S.); geitom@ous-hf.no (G.T.); metsil@ous-hf.no (M.S.B.)

**Keywords:** RBC genotyping, blood groups, blood group antigens, hemoglobinopathy, alloimmunization, blood transfusion

## Abstract

**Background:** Blood transfusion has an essential place in the treatment of several patient groups but entails alloimmunization risk. The provision of phenotype-matched packed red blood cell concentrates (PRBCCs) is recommended for preventing alloimmunization in lifetime transfusion-dependent patients, while antigen-negative PRBCCs are a prerequisite for patients that already have blood group antibodies against clinically significant antigens. A large blood donor corps that is extendedly typed for clinically important red blood cell (RBC) antigens is therefore extremely valuable and can be achieved by high-throughput RBC genotyping. Probability of finding antigen-negative PRBCCs will then increase. **Methods:** RBC genotyping was performed using EDTA-blood. Following DNA extraction, ID CORE XT, a multiplex PCR- and hybridization-based genotype test utilizing Luminex xMAP technology, was employed. The predicted phenotypes were reported. **Results:** In this retrospective study, 92,796 RBC antigens were genotyped in 2508 blood donors, analyzing 48 donor samples in one setup, between 2015 and 2021. These results were compared with results available from serologic phenotyping, and discrepancies were investigated. Antigen-negative blood donors, especially those negative for clinically significant high-frequency antigens, combinations of antigen-negatives and positives for low-frequency antigens, were identified, in addition to variants. **Conclusions:** In the vast majority of discrepancies, repeat serology confirmed the genotype result. Therefore, we conclude that RBC genotyping using ID CORE XT is an effective and accurate method that complements serology. As many donors stop donating blood due to various reasons, we must continue genotyping to maintain and preferably increase the number of extendedly typed blood donors, which are needed for the provision of antigen-negative blood units.

## 1. Introduction

Today, 48 blood groups and 370 blood group antigens in these systems are recognized by the International Society of Blood Transfusion (ISBT) (https://www.isbtweb.org/isbt-working-parties/rcibgt.html, accessed 1 July 2025). Antibodies to clinically important blood group antigens, like ABO, RH1 (RhD), RH2 (C), RH4 (c), RH3 (E), RH5 (e), KEL1 (K), JK1 (Jk^a^), JK2 (Jk^b^), FY1 (Fy^a^), FY2 (Fy^b^), MNS3 (S) and MNS4 (s), can cause acute or delayed hemolytic transfusion reactions [1,2,3]. According to the Norwegian national guidelines for the transfusion services, the above-mentioned antigens should be taken into consideration as a preventive measure when choosing packed red blood cell concentrates (PRBCCs) for patients with life-long transfusion needs due to hematological conditions like sickle cell disease (SCD) and thalassemia [4]. As it is impossible to match all the 370 antigens in an allogeneic blood transfusion setting and difficult to match even only for the clinically significant antigens, alloimmunization may occur in antigen-negative patients and those with partial antigen expression when they receive antigen-positive blood units. Luckily, not all antibodies are clinically significant. Autologous blood transfusion may sometimes be the key. However, not many patients’ clinical conditions will allow an autologous blood donation of 450 mL one or more times to freeze for long-term storage and then thaw to transfuse when needed. Usually, a blood-requiring surgery or treatment of malignant hemato-oncologic diseases requires much more than a couple of blood units, necessitating additional allogenic transfusions. Autologous transfusions are not a viable option for patients with hemoglobinopathies, as their red blood cells (RBCs) are structurally and functionally abnormal. Consequently, PRBCCs from healthy blood donors are needed.

Alloimmunization can be a significant problem in transfusion-dependent patients [5,6]. Patients with hemoglobinopathies, including those with SCD and thalassemia major, may require transfusions every 3–6 weeks throughout their lives: either on a regular basis in patients with thalassemia or during sickle cell crisis and exchange transfusions in patients with SCD. In adult patients with SCD, the alloimmunization rate was reported to be as high as 58% [7,8], and in thalassemia patients, 32.5% [9,10]. Consequently, transfusions with extended phenotype-compatible PRBCCs are recommended as a preventive measure to reduce alloimmunization risk [6,11].

SCD occurs mostly in African populations, while thalassemia is prevalent mainly in people from the Mediterranean countries, the Middle East, the Indian subcontinent and Southeast Asia. In Norway, blood donors are predominantly ethnic Norwegians, and consistent with other Caucasians, they exhibit different antigen frequencies regarding particularly RH2, RH4, FY1, FY2, and JK2 compared to patients with hemoglobinopathies. It is therefore challenging to find antigen-negative blood units for these patients, either as a preventive measure or when they are alloimmunized [5]. The provision of antigen-negative blood units is even more difficult for patients with antibodies to high-frequency blood group antigens (HFA), because HFAs are detected in >90% of all ethnic populations [12,13]. On the other hand, low-frequency blood group antigens (LFAs) are detected in <1% of all ethnic populations [13]. Accordingly, it is not difficult to find PRBCCs for patients with antibodies to LFAs. However, these antibodies are usually not detected because the panel cells used in antibody identification seldom express the corresponding LFAs [14], and may stay undetected until they cause hemolytic transfusion reactions if the antibody is clinically significant. Alloimmunized patients will, for future transfusions, need antigen-negative PRBCCs for the corresponding antibodies [15]. This applies not only to patients with hemoglobinopathies but also to all other patients with clinically significant antibodies; therefore, blood banks need an extended blood group-typed blood donor corps to provide antigen-negative PRBCCs to all these patient groups.

At the Department of Immunology and Transfusion Medicine, Section for Immunohematology, blood donors and patients are routinely ABO- and RH1-typed using serological methods because these methods are easy, practical and cheap, but serology has some limitations. Blood donors are also routinely phenotyped for RH2, RH4, RH3, RH5 and KEL1. Mixed field reactions in patient samples due to recent blood transfusion may be difficult to interpret, and a positive direct antiglobulin test (DAT) may lead to inconclusive results. Not all antigens, like Dombrock blood system antigens, have commercial antisera either. In addition, serology may miss detection of weak antigens or variants [16]. RBC antigens can be genotyped as long as their coding genetic background and DNA polymorphism are known. RBC genotyping has some disadvantages, though, and the most important one is the high cost that prevents extended RBC genotyping from being a routine screening of all blood donors and patients. Because the kits are designed with specific probes only for known alleles, a major drawback of using commercial kits in RBC genotyping is their inability to detect novel mutations and very rare variants. However, serologic phenotyping will also cost at least as much when the same number of antigens are to be typed. Genotyping is time-consuming and therefore will not be suitable for patients requiring urgent transfusions. However, especially for patients requiring lifetime transfusions and for those with several blood group antibodies, samples can be genotyped at the first opportunity, and the results will then be useful in further transfusions. But using high-throughput platforms can counterbalance these disadvantages, since many blood donors can be genotyped for several antigens at one setup. We therefore initiated this study, with high-throughput RBC genotyping, aiming to investigate and identify primarily antigen-negative blood donors, those that are negative for HFAs, special combinations of antigen-negatives and blood donors with LFAs. We have also compared antigen frequencies in our blood donors with previously reported results from the Norwegian population, in addition to Caucasians, Africans and Asians, since patients with hemoglobinopathies are mainly of African and Asian origin.

## 2. Materials and Methods

### 2.1. Study Design and Blood Donors

This retrospective quality assessment study was carried out at the Section for Immunohematology, Department of Immunology and Transfusion Medicine, Oslo University Hospital (OUH), between 2015 and 2021. A total of 2508 blood donors who had donated blood at least twice and had blood groups O and A, both RH1-positive and -negative, except for R^1^R^2^, were included.

### 2.2. Ethics

The study was approved as a quality assurance study by the data protection officer for research at OUH, 22/08243.

### 2.3. Serologic Tests

Upon donor registration, routine testing comprises ABO- and RH1-typing, in addition to an antibody screen. ABO- and RH1-typing are subsequently performed at every donation, whereas the antibody screen is repeated only if, in the intervening time, the donor has received a transfusion and/or has been pregnant, following the mandatory deferral period. An antibody screen using the indirect agglutination test (IAT) at 37 °C on fully automated platforms was performed with three in-house screening cells and LISS/Coombs ID-Cards (BioRad GmbH, Cressier, Switzerland) with the IH-1000 (DiaMed GmbH, Cressier, Switzerland) or Poly Cassettes, anti-IgG, -C3d; polyspecific (Ortho Clinical Diagnostic, Männedorf, Switzerland) with Vision Max (Ortho Clinical Diagnostic, Männedorf, Switzerland). When the antibody screen was positive, antibody identification (IAT) was performed with an in-house panel of 14 blood group O cells of varying phenotypes, using LISS/Coombs ID-Card Bio-Rad. Serologic phenotyping for the main antigens (RH2, RH4, RH3, RH5 and RH8) in the Rh blood group system and KEL1 was performed using the manufacturer’s instructions. Additional phenotyping was performed, if necessary, i.e., when compatible PRBCCs were needed either for patients with the corresponding antibodies or in a preventive antigen-matching setting. Some of the phenotypings were performed almost 25 years ago.

### 2.4. RBC Genotyping

Ethylenediaminetetraacetate (EDTA)-anticoagulated whole blood samples were collected from the blood donors and kept at –20 °C unless DNA extraction was going to be performed within four weeks. DNA extraction was performed using the automated extractor MagnaPure 96 (Roche, Mannheim, Germany) until August 2018 and thereafter using the QIAsymphony-SP (Qiagen GmbH, Hilden, Germany) with the QIAsymphony DSP DNA kit (Qiagen GmbH, Hilden, Germany) in Elution Microtube racks (EMTR). DNA concentration and purity were not tested routinely unless there were problems with the analysis, including “no calls” that may imply too low/high a concentration or bad DNA quality. DNA concentration and purity of the eluate were then measured using NanoDrop spectrophotometry (Thermo Scientific, Foster City, CA, USA). Concentration requirement was 20–80 ng/µL and purity 1.63–2.1. Forty-eight samples were analyzed per kit, with a total of 37 antigens in each sample. All samples were analyzed for the major antigens in the following blood group systems; Rh (RH2, RH4, RH3, RH5, RH8, RH10, RH19, RH20, RH34), Kell (KEL1, KEL2, KEL3, KEL4, KEL6 and KEL7), Kidd (JK1 and JK2), Duffy (FY1 and FY2), MNS (MNS1, MNS2, MNS3, MNS4, MNS5 and MNS7), Diego (DI1 and DI2), Dombrock (DO1, DO2, DO4 and DO5), Colton (CO1 and CO2), Yt (YT1 and YT2) and Lutheran (LU1 and LU2).

After the automated DNA extraction, the ID CORE XT kit (Grifols Biopharma, Barcelona, Spain) that utilizes Luminex xMAP technology (Luminex Corporation, Austin, TA, USA) was used following the manufacturer’s instructions. Analysis with the ID CORE XT kit consists of four steps: amplification, hybridization, labeling and quantification. The DNA samples were amplified and biotinylated by multiplex PCR using Thermo Cycler Veriti (Applied Biosystems, Foster City, CA, USA). The PCR products were denatured and hybridized to oligonucleotide probes coupled with color-coded beads. Hybridized DNA was labeled with a fluorescent conjugate, and the resulting signals were detected with Luminex 200 (Bio-Rad, Hercules, CA, USA). The raw data were processed with the ID CORE XT Analysis Software BIDS XT (until 2017 software version 1.6, and then version 1.8) to obtain genotypes and predicted phenotypes, and finally, predicted phenotypes were automatically transferred to the Laboratory Information Management System (LIMS) ProSang (Omda, Oslo, Norway).

The predicted phenotypes were positive (normal antigen expression), negative (no antigen expression), no call (NC) (i.e., the software was not able to assign a call for one or more polymorphisms due to low signal intensity or low bead count and/or indeterminate genotype), and unknown (i.e., that particular combination of single nucleotide polymorphism (SNP) is not included in the analysis software or has not been previously described as associated with a phenotype). Samples with “no calls” for only one or two genotypes/predicted phenotypes and samples with “unknown” results were not reanalyzed. In the routine, serological typing results, when available, were used instead, but serological results were not used to replace genotypes in statistics for this study. The ID CORE XT results include explanatory notes for some outputs if antigen expression is altered (weak, partial or variable). We did not perform sequencing to confirm our findings when we suspected or detected variants.

Genotype and phenotype results were compared for the antigens we could phenotype serologically, and discordant results were investigated. A discrepancy was defined as either positive serology results with a negative genotype result or vice versa. When a discrepancy was revealed, our intention was to repeat the genotyping and/or serology using either the same or a new sample, if available. If there was still no explanation, genotyping was also performed using RBC-FluoGene vERYfy and Rare kits (inno-train, Kronberg, Germany) and/or the sample was shipped to and analyzed by Grifols.

### 2.5. Statistical Analysis

Population frequency comparisons, seen in Table 1 (our study group compared with Caucasians, Africans, and Asians), for each antigen to evaluate the strength of the observed differences were performed using Chi-square tests for independence (for adequate expected frequencies) and Fisher’s exact test (for small expected frequencies). Anthropic Claude Sonnet 4.5 (San Francisco, CA, USA) was used for these statistics. Statistical significance was defined as *p* < 0.05. Missing data were excluded from comparisons.

Probabilistic Rh haplotype frequencies (Table 2) were estimated using Excel and the Solver add-in (GRG Nonlinear algorithm) by minimizing the sum of squared errors (SSE) between the observed and modeled phenotype proportions. To evaluate the internal consistency of RH1, RH2 and RH3 phenotypes, we estimated Rh haplotype frequencies using a probabilistic model under Hardy-Weinberg equilibrium assumptions. The analysis combined serologic RH1 typing with RHCE genotyping data from all 2501 blood donors (excluded the seven “no calls” for each Rh antigen) and classified according to RH2 and RH3 genotype combinations. All frequencies were constrained to be non-negative and to sum to one. Confidence intervals (95%) for each haplotype frequency were calculated using binomial approximations (f ± 1.96 √ [f (1 − f)/(2*N*)]), where *N* = 2501 donors.

Antigen frequencies are calculated by dividing the number of positive results by the total number of blood donors genotyped after excluding “no calls” and “unknown” results from the total number (denominator) for each antigen. Frequency percentages of antigen-negative combinations (Table 3) were calculated by dividing the number of negative results of two antigens (both antigens are negative) by the total number of blood donors typed.

## 3. Results

In total 92,796 RBC antigens were genotyped in 2508 blood donors, 37 antigens in every donor. When the study was finalized, 22.4% (*N* = 563) of the blood donors were not active anymore. 45.5 percent of the blood donors had blood group A (*N* = 1175 donors), and 54.5% had blood group O (*N* = 1397 donors). Almost 76 percent were RH1 positive (*N* = 1899) and 25% were RH1 negative (*N* = 609). The genotyping results for all donors are shown and compared with published frequencies of genotypes in Caucasian, African and Asian populations in Table 1.

**Table 1 biomedicines-13-02907-t001:** Antigen frequencies in our blood donors, genotyped (except for RH1 that was typed serologically) using ID CORE XT, compared to data from Caucasians, Africans and Asians [12,17].

Blood Group System	Antigen	Frequency in the Study Group (%)	Frequency in Caucasians (%)	Comparison Between the Study Group and Caucasians (*p* Value)	Frequency in Africans (%)	Comparison Between the Study Group and Africans (*p* Value)	Frequency in Asians (%)	Comparison Between the Study Group and Asians (*p* Value)
**Rh**	**RH1**	75.7	85	<0.0001	92	<0.0001	99	<0.0001
	**RH2**	57.9	68	<0.0001	27	<0.0001	93	<0.0001
	**RH4**	79	80	0.58	98	<0.0001	47	<0.0001
	**RH3**	18.2	29	<0.0001	22	0.03	39	<0.0001
	**RH5**	97	98	0.20	98	0.20	96	0.27
	**RH8**	2.7	2	0.38	1	0.01	-	N/A
	**RH10**	0.28	1	0.10	30	<0.0001	-	N/A
	**RH19**	97.7	98	0.76	98	0.76	98	0.76
	**RH20**	0.24	<0.01	0.50	26–40	<0.0001	<0.01	0.50
	**RH34**	97	98	0.16	97	0.99	-	N/A

**Kell**	**KEL1**	7.9	9	0.47	2	<0.0001	Rare	N/A
	**KEL2**	99.7	99.8	0.99	100	0.25	-	N/A
	**KEL3**	2.7	2	0.38	<0.01	<0.0001	-	N/A
	**KEL4**	100	100	0.99	100	0.99	100	0.99
	**KEL6**	0.08	<0.01	0.07	20	<0.0001	-	N/A
	**KEL7**	100	100	0.99	99	0.99	-	N/A

**Kidd**	**JK1**	76.6	77	0.90	92	<0.0001	72	0.48
	**JK2**	74.8	74	0.64	49	<0.0001	76	0.64

**Duffy**	**FY1**	68.4	66	0.24	10	<0.0001	99 †	<0.0001
	**FY2**	80.5	83	0.13	23	<0.0001	18.5 ††	<0.0001

**MNS**	**MNS1**	80.5	78	0.21	74	<0.0001	-	N/A
	**MNS2**	69.2	72	0.20	75	0.01	-	N/A
	**MNS3**	52.5	55	0.28	31	<0.0001	-	N/A
	**MNS4**	90.6	89	0.24	93	0.07	-	N/A
	**MNS5**	100	99.9	0.99	99	0.90	-	N/A
	**MNS7**	0.04	<0.01	0.99	-	N/A	rare	N/A

**Diego**	**DI1**	0.12	<0.01	0.99	<0.01	0.99	<0.01	0.99
	**DI2**	100	100	0.99	100	0.99	100	0.99

**Dombrock**	**DO1**	59.8	67	0.99	55	0.03	12–30	0.99
	**DO2**	87	82	<0.0001	89	0.21	-	N/A
	**DO4**	100	100	0.99	>99	0.99	-	N/A
	**DO5**	100	100	0.99	>99	0.99	-	N/A

**Colton**	**CO1**	99.8	99.5	0.45	99.5	0.45	99.5	0.45
	**CO2**	7.7	10	0.07	10	0.07	10	0.07

**Yt**	**YT1**	99.9	>99.8	0.99	>99.8	0.99	>99.8	0.99
	**YT2**	5.5	8	0.03	-	N/A	-	N/A

**Lutheran**	**LU1**	7	8	0.44	5	0.07	-	N/A
	**LU2**	99.8	99.8	0.99	99.8	0.99	99.8	0.99

†: Thai: 97%. ††: Chinese: 9.2%, Thai: 31%.

Determination of the genotype failed for 123 antigens (0.13%), giving “no calls” in 33 donors. Sixty-three samples had “no calls” for antigens of the Rh (seven “no calls” for each of RH2, RH4, RH3, RH5, RH8, RH10, RH19, RH20, RH34), four for MNS (one “no call” for each of MNS3, MNS4, MNS5 and MNS7), four for Dombrock (one “no call” for each of DO1, DO2, DO4 and DO5), four for Yt (two “no calls” for YT1, two for YT2) and forty-eight for Lutheran (24 “no calls” for LU1, 24 for LU2) blood group systems. One of the sources we refer to [17] states that the antigen frequencies given for Caucasian populations (northern European) are the best studied group by hemagglutination, and that the information was obtained from original publications, Blood Groups in Man, and AABB Technical Manual. Therefore, we assume that the data, and probably also for Africans and Asians, originate from phenotyping.

The frequencies of the Rh haplotypes in our blood donor corps compared with the frequencies in Norwegians, Africans and Asians are shown in Table 2. The Solver optimization converged successfully (SSE = 0.0086), indicating an excellent fit between observed and modeled RhD–C–E phenotype proportions. The estimated haplotype frequencies were r (dce) = 0.49, R^1^ (DCe) = 0.39, R^2^ (DcE) = 0.10, and R^0^ (Dce) = 0.02, with negligible contributions from R^z^, r′, and r″. This distribution is consistent with data reported for Northern European blood donor populations, where r and R^1^ are the dominant haplotypes. The model reproduced the observed group proportions with small residuals (Δ < 7% for all groups), confirming the robustness of the fit. Probabilistic estimation of Rh haplotype frequencies confirmed internal consistency between observed and modeled RhD–C–E distributions (Appendix A).

**Table 2 biomedicines-13-02907-t002:** Probabilistically estimated frequencies of Rh haplotypes in our blood donors (based on serologic RH1 type and RHCE genotypes) compared to the frequencies in Norwegians [18], Caucasians, Africans and Asians [17].

Haplotype	Estimated Frequency in the Study Group (%)	Frequency in Norwegians (%)	Frequency in Caucasians (%)	Frequency in Blacks (%)	Frequency in Asians (%)
**R^1^**	38.9	39.3	42	17	70
**R^2^**	9.8	15.2	14	11	21
**R^0^**	2.3	2.0	4	44	3
**R^z^**	0	Very rare	0	0	1
**r**	49	40.4	37	26	3
**r’**	0	0.9	2	2	2
**r**″	0	0.7	1	0	0

The highest frequencies (≥97%) were seen for the following antigens: RH5, RH19, RH34, KEL2, KEL4, KEL7, MNS5, DI2, DO4, DO5, CO1, YT1 and LU2, while RH4 (79%), FY2 (80.5%), MNS1 (80.5%) and MNS4 (90.6%) were less frequent. The number of blood donors negative for the high-frequency and positive for low-frequency antigens are shown in Table 4 and Table 5, respectively. RH8 was the low-frequency antigen with the highest frequency.

**Table 3 biomedicines-13-02907-t003:** Frequency percentage of antigen-negative combinations.

	RH1-	RH2-	RH4-	RH3-	RH5-	RH8-	KEL1-	FY1-	FY2-	JK1-	JK2-	MNS3-	MNS4-
**RH1-**	-	23.8	0.04	23.6	0.08	24.2	22.4	7.5	5.1	6.6	6.1	11.5	2.5
**RH2-**	23.8	-	0	24.8	3	42.1	38.9	12.9	8.9	10.9	10.6	20.1	4.2
**RH4-**	0.04	0	-	21	0	19.5	19.5	6.8	4.1	6	5	9.9	1.7
**RH3-**	23.6	24.8	21	-	0	79.4	75.2	26.3	15.6	21.8	20.9	38.9	7.2
**RH5-**	0.08	3	0	0	-	3	2.8	0.6	0.8	0.8	0.8	1.6	0.3
**RH8-**	24,2	42.1	19.5	79.4	3	-	89.7	30.6	19.1	25.9	24.5	46.1	8.8
**KEL1-**	22.4	38.9	19.5	75.2	2.8	89.7	-	29.1	18.2	24.2	23.2	43.8	8.2
**FY1-**	7.5	12.9	6.8	26.3	0.6	30.6	29.1	-	0.08	8.2	8.6	15.2	2.8
**FY2-**	5.1	8.9	4.1	15.6	0.8	19.1	18.2	0.08	-	4.8	4.7	9.9	2.2
**JK1-**	6.6	10.9	6	21.8	0.8	25.9	24.2	8.2	4.8	-	0	11.8	2.8
**JK2-**	6.1	10.6	5	20.9	0.8	24.5	23.2	8.6	4.7	0	-	11.4	2.1
**MNS-**	11.5	20.1	9.9	38.9	1.6	46.1	43.8	15.2	9.9	11.8	11.4	-	0
**MNS4-**	2.5	4.2	1.7	7.2	0.3	8.8	8.2	2.8	2.22	2.8	2.1	0	-

**Table 4 biomedicines-13-02907-t004:** Blood donors negative for high-frequency antigens.

High-Frequency Antigen	Number of Negatives	Frequency of Negatives (%)
**KEL2**	7	0.28
**KEL4**	0	0
**KEL7**	0	0
**DI2**	0	0
**DO4**	0	0
**DO5**	0	0
**CO1**	6	0.24
**RH5**	76	3.03
**MNS5**	0	0

**Table 5 biomedicines-13-02907-t005:** Blood donors positive for low-frequency antigens.

Low-Frequency Antigen	Number of Positives	Frequency (%)
**MNS7**	1	0.04
**RH20**	6	0.24
**RH8**	69	2.71

The 95 genotype variants detected by ID CORE XT in 90 blood donors are shown in Table 6. The most common antigen variant was *FY*B*, *FY*B[265T]_FY*X*, followed by *FY*A*, *FY*B[265T]_FY*X*.

Combinations of antigen-negative frequencies are shown in Table 3. The highest percentage was RH8- KEL1- with 89.1%, and then RH8- RH3- and KEL1- RH3-. None of the blood donors were RH1- RH4- or RH4- RH5-.

Discordances between genotype and phenotype were revealed in 30 blood donors (Table 7), mainly (*n* = 16, 51.6%) in the Duffy blood group system. No variants were detected as an explanation of the discordances. In two samples (sample numbers 6 and 14), serologic results were used for the conclusion. Only sample 7 was sent to Grifols for further investigation. The turnaround time for receiving the result was four weeks.

## 4. Discussion

The RBC-genotyped blood donors (*n* = 2573) in this study accounted for approximately 15% of our whole blood donor corps. Blood donors with blood groups O and A were chosen for RBC genotyping, as almost 90% of the Norwegian population, and presumably also the patients, have these blood groups [18]. The ID CORE XT test does not include ABO and RH1 genotyping. We therefore used serologic ABO and RH1 results in our study. R^1^R^2^ blood donors were excluded since they are positive for the main antigens (RH1, RH2, RH4, RH3, RH5) in the Rh blood group system and are not relevant since we need antigen-negative PRBCCs and prioritize the abovementioned antigens as a minimum requirement of phenotype-compatibility. Our strategy of RBC genotyping after at least two successful donations was based on the assumption that the donors would remain active in the donor pool. However, when the study was finalized, 629 donors (24.4%) were deferred due to medical conditions, or because they had moved out of our district or wanted to stop donating for other reasons. The main goal is to have a donor pool where almost all are RBC-genotyped. However, achieving this goal is costly and requires organizational commitment and resources. It will also take time unless the capacity is increased considerably. We cannot expect our donors to commit to a lifetime donor career. Thus, we need to accept that RBC genotyping of the deferred blood donors has economic consequences without significant benefit for the patients. Recently, we have also initiated RBC genotyping of blood groups other than O and A, expecting that the demand/pressure on O donors, especially, will be reduced, since we can now choose and issue phenotype-compatible B and AB blood units to patients with these blood groups.

High-throughput RBC genotyping with ID CORE XT is an effective way of analyzing 48 donor samples in one setup. Initially, we had several “no calls” (0.14%). Troubleshooting finally identified that the wrong type of thermal block, which led to suboptimal amplification and biotinylation during multiplex PCR, was installed in the thermocycler. After the correct type was installed, there were almost no “no calls”. We decided not to re-analyze these samples, since it would be too costly to repeat genotyping, and used instead the serologic results available.

Regarding Rh haplotypes, a probabilistic estimation of Rh haplotype frequencies was performed. The analysis confirmed that the observed CDE genotype distribution is internally consistent and well described by a Hardy-Weinberg equilibrium model (SSE = 0.0086). The estimated haplotype frequencies, dominated by r and R^1^, were highly concordant with published values for Northern European populations [17]. The overrepresentation of RH1-negative donors in our dataset may reflect selection bias due to higher demand from the blood bank, retention and/or higher commitment among such donors, rather than a true genetic deviation. The absence of rare haplotypes (R^z^, r′, r″) is expected given their extremely low prevalence (<0.1%) in Caucasian populations. By integrating probabilistic haplotype modeling, we have provided independent confirmation of the accuracy of the genotyping results and the genetic representativeness of our donor cohort, thereby providing a population-based statistical validation. The highest probabilistically estimated frequencies in our blood donors were r and R^1^, followed by R^2^. We cannot distinguish between RH1/RH1 and RH1/RH1- without zygosity analysis, which is not available at our laboratory.

Tordon et al. identified several antigen-negative rare donors (five KEL2-, four MNS5-, two JK (1–2-) and two RH1- -) using selected pheno- and genotyping based on race and ethnicity [19]. In Norway, unlike the Canadian strategy, we neither ask nor record the donors’ race/ethnicity. However, doing so would be a much more targeted strategy. Although the ID CORE XT kit detects many of the most clinically important antigens, including those typical for Africans and Asians, it does not cover ABO, RH1, including variants, and other clinically significant antigens such as VEL, DI3 and JR1. The absence of routine screening for these latter clinically significant antigens may pose risks in patients, especially in those with rare blood types. There are commercial kits available for genotyping ABO, RH1 (also variants) and several low-frequency antigens. However, including more and separate commercial kits in the routine increases the costs. Not all these kits are suitable for high-throughput testing, either. Nevertheless, there are possibilities with Next-Generation Sequencing (NGS) and efforts, where not only RBC antigens but also platelet, leukocyte and neutrophil antigens can be genotyped. Nevertheless, Norwegian national guidelines for transfusion services do not require donor testing and preventive antigen-matching for low-frequency antigens. Rare phenotypes as DO4-, DO5-, KEL7- and MNS5- are more common in Africans [12], while 100% of the blood donors in our study were DO4+, DO5+, KEL7+, MNS5+, KEL4+ and DI2+. Antigen-negativity for high-frequency antigens like KEL4, VEL and KEL2 is also rare in Caucasians; only 0.02% of Norwegians are KEL4- and 0.2% KEL2- [18]. We identified, though, seven KEL2- (0.28%) among our blood donors. Only 3.03% was RH5-. RH19 is highly frequent in all populations. The only blood donor that was RH19- was identified with the *RHCE*ce[712G]* genotype.

Only seven of our blood donors were positive for low-frequency antigens; one MNS7+ and six RH20+. The standard antibody identification panels do not include MNS7+ and RH20+ cells; thus, anti-MNS7 and -RH20 will not be recognized in routine immunohematological investigations unless they cause hemolytic transfusion reactions. These antibodies are much more frequent in Southeast Asia [20,21]. Therefore, information about the ethnic origin of donors and patients can guide antibody investigation. Samples from these donors are included in our diagnostic biobank to be used in investigations when anti-MNS7 and anti-RH20 are suspected.

*FY*B, FY*B[265T]_FY*X*, which is expressed on the RBCs of 1–2% of Caucasians, comprised the largest group of variants in the current study. *FY*A*, *FY*B[265T]_FY*X* and *FY*B_GATA* were the next-most frequent variants. Lopez et al. reported *FY*A*, *FY*B[265T]_FY*X* as the most frequent variant in their study, not only in donors but also in patients and neonates [22]. Sheppard et al., on the other hand, found GATA mutation affecting the FY2 expression in almost 80% of the patients with SCD [23]. *FY*B*_GATA indicates that the FY2 antigen is silenced by the GATA mutation (FY*-67t >c) that prevents FY2 expression only on RBCs and leads to Fy^null^ phenotype [24]. These individuals do not develop anti-FY2 because the FY2 antigen is expressed on other cells/tissues. [265T]_*FY*X* leads to the weak expression of the FY2 antigen on RBCs and can be missed in serologic typing, giving false negative results [24]. There will be no clinical consequences of false negative phenotyping in a patient since the patient will then receive antigen-negative PRBCCs, but false negative results in a blood donor sample may immunize an antigen-negative patient and cause hemolytic transfusion reaction if the patient has a clinically significant antibody to the corresponding antigen.

Although we initially assumed that our blood donors were mainly ethnic Norwegians, the occurrence of certain variants like *FY*B_GATA and FY*A, FY*B_GATA* together with one MNS7+ and six RH20+ donors indicates that we also have donors of other ethnicities.

Anti-RH3, -RH2 and -KEL1 are the most frequent antibodies detected in patients with SCD and thalassemia [25,26,27]. Phenotype-compatible blood transfusions, taking into consideration the main antigens in the Rh blood group system and KEL1, will prevent and reduce the alloimmunization rate [6,25]. Alloimmunization risk will be further reduced by providing extended phenotype-matched PRBCCs, taking more antigens (JK1, JK2, FY1, FY2, MNS3 and MNS4) into consideration. However, finding such units is not easy for these patients with long-term transfusion needs [6]. We have shown that less than 5% of our blood donors have the following antigen-negative combinations: RH2- RH5-, RH2- MNS4-, KEL1- RH5-, and RH1- MNS4-, while RH1- FY2- was 5.1%. More frequent combinations, yet still lower than 9% are RH2- FY2-, RH3- MNS4-, KEL1- MNS4-, RH1- FY1-, RH1- JK1-, and RH1- JK2-.

Since the ethnic origin of patients with SCD and thalassemia is different than that of most Norwegian blood donors, we compared the antigen frequencies of our blood donors with previously reported frequencies in Africans and Asians to illustrate the possibility of finding antigen-negative PRBCCs in our stocks. We could confirm differences between Africans and our blood donor population regarding the following antigens: RH2, RH10, RH20, KEL1, JK1, JK2, FY1, FY2 and MNS4. For example, Africans have 30% lower frequency of RH2 antigen and up to 40% higher frequencies of RH10 and RH20 antigens, the latter two being true low-frequency antigens in Caucasians [17]. This also applies to our blood donors (<0.3%). Partial/weak RH2 and RH5 are associated with RH20, and are common in Africans, while this is not the case in Caucasians [28,29]. RH2- RH5- PRBCCs are recommended to patients with partial/weak RH2 and RH5, as they can develop anti-RH2 and anti-RH5 [17]. It will not be difficult to find RH2- PRBCCs for RH2- patients and patients with partial/weak RH2 antigen since 42% of our blood donors were RH2-. Almost 80% of Africans are RH3-. It is more difficult to find PRBCCs for patients with partial/weak RH5 due to the high frequency of RH5. In addition, transfusing RH5- PRBCCs may entail alloimmunization to RH3, as it is almost impossible to retrieve both RH3- RH5- units [17]. Africans have a higher frequency of R^0^R^0^ (RH1+ RH2- RH3-) compared to the Norwegian population: 45.8% versus 0.04, respectively [17,18]. R^0^R^0^ patients can also be transfused with rr (RH1- RH2- RH3-) PRBCCs to avoid RH2 and RH3 alloimmunization, but only 25% of our blood donors are RH1 negative, while the frequency in the Norwegian population is 15%. Therefore, the absolute requirement of ABO compatibility, together with recommendations for phenotype compatibility for other clinically significant antigens, will limit the number of PRBCCs we can provide for R^0^R^0^ patients.

Approximately 68% of Africans are Fy^null^, while Fy^null^ Caucasians barely exist, and that makes finding Fy^null^ blood donors for Fy^null^ patients almost impossible. As mentioned above, when the homozygote *FY*B* allele carries a GATA mutation, Fy^null^ patients will not develop anti-FY2. Accordingly, choosing only FY1- PRBCCs for Fy^null^ patients prophylactically is recommended [4,6]. Having almost 32% FY1- in our blood donor corps enables us to find phenotype-matched PRBCCs for these patients. RH1- in combination with other antigens was less than 23.8%. Regarding some other combinations, less than 10% of our blood donors had the antigen-negative combinations of, for example, RH2- FY2-, RH4- FY1-, RH4- JK1-, RH4- MNS3- and FY2- MNS3-, and there was an even lower frequency of RH4- FY2-, RH4- JK2-, RH4- MNS4-, JK1- FY2-, JK2- FY2- and FY2- MNS4-. It is therefore challenging to find matched PRBCCs for antigen-negative patients with corresponding antibodies and as a preventive measure for patients without antibodies.

Comparing our blood donors with Asians showed that our blood donor population had a higher frequency of RH4, KEL1 and FY2. Since 99% of Asians are RH1 positive, provision of RH1-matched PRBCCs for patients with Asian ethnicity will not be demanding at all, considering that 75.8% of the blood donors in our study were RH1 positive.

Hyperhemolysis in patients with SCD and thalassemia may be life-threatening. Both the transfused and the patient’s own RBCs will then be hemolyzed, resulting in a drastic fall in hemoglobin, below pre-transfusion level. The index antibody cannot always be identified, but Dombrock antibodies are shown to cause severe acute and/or delayed hemolytic transfusion reactions [30]. DO1 and DO2 are polymorphic antigens with frequencies of almost 60 and 87%, respectively, in our blood donors. These frequencies are generally comparable to Caucasians and Africans. However, the frequency of DO1 is significantly lower in Asians, suggesting that transfusing DO1-negative patients with thalassemia may pose an alloimmunization risk. Anti-DO1 and anti-DO2 are often found in combination with other antibodies and can be difficult to detect, as Dombrock antibodies tend to degrade in vitro and weaken in vivo [31,32]. Nevertheless, there is no recommendation for preventive Dombrock antigen-negative PRBCCs for patients with hemoglobinopathy [7,11].

In total, there were 30 discrepancies between the results of serology and genotyping with ID CORE XT in 30 blood donors. Sixteen of the discrepancies were in the Duffy blood group system. In 12 of the 16 cases, the historical FY1 phenotype was positive, using a manual method performed nearly 25 years ago, while genotyping was negative, and no variants were detected in these blood donors. Ten of these phenotypings were repeated in new samples, and all 10 were negative. We could not find any explanation for these former erroneous serologic results. Two other discrepancies were due to wrong blood in tube. Another donor (sample 7) was initially tested RH2-negative by genotyping, despite a positive serological result. Subsequent sequencing conducted by the distributor (Grifols) confirmed the RH2-positivity. The C-negative result was attributed to a 109 bp insertion within intron 2, which coincided with the probe site in the *RHCE* allele. This discrepancy illustrates the limitations of kit-based RBC genotyping. In one other donor sample (sample 2), RH2 genotyping was registered as negative, while serology showed a positive result. This genotype result should not have been validated in the first place, because a “no call” was overlooked. Repeated genotyping using the FluoGene platform showed a positive result. In sample 6 (Table 7), despite negative serology (FY2-), genotyping was positive. Genotyping was not repeated, but serology was repeated twice and was still negative. In ProSang (LIMS), regardless of the method, the last two consistent results override the former result(s). Therefore, the conclusion was negative, i.e., the same as the serological result. This discrepancy could be due to a variant that the kit did not detect and could have been revealed by sequencing. It is plausible that such a variant may cause weak expression of the FY2 antigen, which can be missed by serology. A similar situation regarding ProSang also applies to sample 14, which was RH2+ by serology and RH2- by genotyping. Serology was repeated twice after the discrepancy was revealed, which again overrode the result of genotyping. Sequencing could likely have resolved this discrepancy but was not performed. However, overall, repeated phenotyping (in 20 of 30 cases) mainly confirmed genotyping results. Serology, to start with and as a complementary test, is essential and has been performed for each discrepancy case except for sample 10 (Table 7). While investigating discrepancies, only four samples were analyzed using the other genotyping platform (FluoGene) available at our laboratory. We need to be more proactive in using FluoGene in discrepancy investigations.

The strength of the current study is the large sample size and the design of the study, with the possibility of comparing genotyping results with serology. A major drawback is the absence of RH1 genotyping of donors, especially those that are RH1-negative by serology, like DEL individuals, but may potentially RH1-alloimmunize an RH1-negative patient. However, economic concerns and personnel resources have been the limiting factors. RH1 genotyping of 2508 blood donors using a complementary technique would have cost significant amounts. On the other hand, blood donors are considered RH1-positive regardless of how weak the serological typing result is. Blood units from such a donor will not be transfused to an RH1-negative patient. Another limitation of our study was the lack of sequencing when investigating the discrepancies and the samples with rare variants to confirm our findings. In addition, the retrospective nature of our study may have contributed to missing data, despite consistent efforts to follow up and resolve discrepancies. Donors were not contacted solely for control sampling to minimize donor inconvenience. Samples were collected instead at their subsequent routine donation. This occasionally resulted in considerable delays when donors were not available; these donors were temporarily, or in some cases permanently, deferred. Outsourcing further sample investigation, like sending samples to Grifols also leads to delays in elucidating the discrepancies due to the turnaround time of approximately four weeks, although it costs only shipping expenses. Until the conclusion is available, the donor is considered “not typed” for the antigen, so as not to risk alloimmunization in an antigen-negative patient. Although it is seldom necessary (only one sample in our study), dependency on an external facility is still a drawback.

## 5. Conclusions

Routine ABO/RH1 typing will likely continue using serological methods for many years to come, especially in an emergency setting, because it is fast, affordable and usually conclusive. On the other hand, despite being more time-consuming and costly, RBC genotyping, especially by means of high-throughput methods, has the great advantage of extended typing many blood donors in one setup. ID CORE XT has, however, several limitations, including restricted applicability to several clinically significant antigens (e.g., VEL, DI3 and JR1), ABO and especially RH1. Failure to detect variants in donors may potentially lead to RH1-alloimmunization in RH1-negative patients. Like other commercial kits, ID CORE XT does not detect novel mutations and very rare variants due to its design with specific probes only for known alleles. Sequencing is necessary to resolve discrepancies/inconclusive results and confirm variants. RBC genotyping has the potential to improve the provision of compatible PRBCCs for patients with hemoglobinopathies and for those with clinically significant antibodies. Nevertheless, in this internal quality assurance study running over seven years, the number of extended RBC-genotyped donors (*n* = 2508), approximately 358 annually, is suboptimal. By the end of the study, 22.4% (*n* = 563) of the blood donors were either deferred because of medical conditions or moved out of Oslo or wanted to stop donating. We need, therefore, to continue RBC genotyping to maintain and increase the number of genotyped blood donors.

## Figures and Tables

**Table 6 biomedicines-13-02907-t006:** Variants in 90 blood donors at OUH.

Blood Group System	Genotype Results	Number of Samples (%)
**Rh**	*RHCE*ce, RHCE*ce[712G]*	1 (0.04)
	*RHCE*ce, RHCE*c[733G]*	5 (0.2)
	*RHCE*Ce, RHCE*ce[733G]*(36)	1 (0.04)
	*RHCE*Ce, RHCE*ceAR*(14)	1 (0.04)
	*RHCE*cE, RHCE*ce[733G]*	5 (0.2)
**Kidd**	*JK*A, JK*B_null(871C)*	1 (0.04)
	*JK*B, JK*B_null(871C)*	1 (0.04)
	*JK*A, JK*B_null(IVS5–1a)*(31)	1 (0.04)
**Duffy**	*FY*A, FY*B[265T]_FY*X* (32)	16 (0.64)
	*FY*B, FY*B[265T]_FY*X*	26 (1.04)
	*FY*B_GATA*	13 (0.52)
	*FY*A, FY*B_GATA*(1)	7 (0.28)
	*FY*B, FY*B_GATA*	11 (0.44)
**MNS**	*GYPB*S, GYP.Mur*	1 (0.04)

The following variants were co-detected in the same blood donors: RHCE*Ce, RHCE*ceAR(14) and FY*B, FY*B_GATA RHCE*Ce, RHCE*ce[733G](36) and FY*B, FY*B_GATA. RHCE*ce, RHCE*ce[733G] and FY*B_GATA. RHCE*ce, RHCE*ce[733G] and FY*B, FY*B_GATA. Footnotes from ID Core^XT^ User Manual: (1): Also possible, although less likely: FY*A_GATA, FY*B. Possible phenotype: Fya- Fyb+. (14) Also possible, although less likely: RHCE*ce, RHCE*CeVG or RHCE*ce[733G], RHCE*CeFV or RHCE*ce[712G], RHCE*Ce[733G]. Possible phenotype: C + weak and V and vs. unknown prediction. (31): Also possible, although less likely: JK*A_null (IVS5–1a), JK*B. Possible phenotype: Jka- Jkb+. (32): Also possible, although less likely: FY*B, FY*A[265T]_FY*X. Possible phenotype: Fya+^w^ Fyb+. (36) Also possible, although less likely: RHCE*ce, RHCE*Ce[733G]. Possible phenotype: c+, V and vs. Unknown.

**Table 7 biomedicines-13-02907-t007:** Overview of discrepancies in total of 30 consecutive samples.

Donor Sample Number	Antigen	Serology	ID Core XT	Repeat ID Core XT	Repeat Serology	FluoGene	Progenika/Grifols	Conclusion
1	**CO1**	+	-		-			-
2	**RH2**	+	- ^1^	+	+	+		+
3	**CO2**	-	+		-/+ ^2^			+
4	**LU1**	-	+		+			+
5	**JK2**	+	-		-/- ^3^			-
6	**FY2**	-/- ^4^	+		-/- ^5^			-
7	**RH2**	+	-		+/+ ^6^	+	+	+
8	**FY2**	-	+		-			?
9	**JK1**	+/- ^7^	+		+			+
10	**CO2**	-	+			+		+
11	**RH8**	+	-		-	-		-
12	**FY2**	+	-		-			-
13	**JK1**	-	+		+			+
14	**RH2**	+	-		+/+ ^8^			+
15	**KEL2**	-	+		+			+
16	**MNS1**	+	-		-			-

Additionally, 12 discrepancies were related to FY1. In two other separate samples (not part of the 12 previously mentioned, but additional discrepancies), there was wrong blood in tube. ^1^: Further examination showed a negative result on the borderline with “no call values”. ^2^: two separate phenotypings one year apart. The first one with a patient plasma with previously identified and confirmed anti-CO2, the second with a commercial reagent. ^3^: Two separate phenotypings one year apart. ^4^: Two separate phenotypings three years apart. ^5^: Two separate phenotypings one year apart. ^6^: Two separate phenotypings one year apart. ?: Blood donor was not available for a confirmatory test. ^7^: Two separate phenotypings five years apart. ^8^: Two separate phenotypings one year apart.

## Data Availability

The original contributions presented in this study are included in the article. Further inquiries can be directed to the corresponding author.

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
