# Peer review of "Blood Donor RBC Genotyping at the National Advisory Unit on Immunohematology at Oslo University Hospital"

_biomedicines, 2025, doi:10.3390/biomedicines13122907_

Round 1
Reviewer 1 Report
Comments and Suggestions for Authors
Retrospective study (2015–2021) on 2,573 Norwegian donors genotyped for 37 red blood cell antigens using ID CORE XT (Luminex), with targeted comparison to historical serological phenotyping.
Key message from the authors:
The platform is efficient and overall concordant with serology. It is a useful tool to establish an extended donor pool to meet the needs of transfusion-dependent patients (e.g., SCD, thalassemia). The authors therefore advocate for continued and expanded donor genotyping efforts.
Strengths of the study:
- Large donor cohort and high number of antigens tested, with a long recruitment period providing insights into real-world logistics.
- Direct genotype/phenotype comparison.
- Transfusion relevance: focus on clinically critical antigens (Rh, Kell, Kidd, Duffy, MNS, etc.).
- Population-level perspective.
However, there are major limitations:
- ISBT nomenclature should be used consistently throughout the manuscript.
- Population frequency comparisons require more sources and should specify whether data originate from genotyping or phenotyping. No statistical analyses are provided to evaluate the strength of the observed differences.
- Lack of access to ethnic background data limits interpretation. A probabilistic analysis based on Rh haplotypes could have been performed.
- Management of no calls: samples were not retested for economic reasons, which may bias population-level distributions. Why not exclude the donor’s entire genotype when a no call occurred for one antigen?
- These technical issues highlight the importance of complementary serological testing, especially when only one genotyping method is used without sequencing confirmation.
- Variants detected in 91 donors were not confirmed with a second method (e.g., sequencing, another commercial kit, Fluogene, or in-house PCR).
- For variants (Table 5), an additional column would be useful to indicate transfusion impact and how such donors are handled in product selection.
- ABO genotyping was not performed (not a limitation). However, the absence of RhD genotyping is problematic given the existence of variants with clinically significant transfusion impact. A complementary technique (e.g., Fluogene or similar) would have been appropriate, and would in any case be necessary if this strategy is to be deployed.
Minor limitations:
- Ensure consistency across tables.
- The organizational and economic impact remains insufficiently addressed; such a study would be highly relevant for drawing broader conclusions in this area.
Author Response
For research article
|
Response to Reviewer Comments
|
||
|
Summary |
|
|
|
Thank you very much for taking the time to review this manuscript. Please find the detailed responses below and the corresponding revisions/corrections in track changes in the re-submitted files. |
||
|
Point-by-point response to Comments and Suggestions for Authors |
||
|
Comments 1: ISBT nomenclature should be used consistently throughout the manuscript. |
||
|
Response 1: Thank you for pointing this out. We have now revised our manuscript using ISBT nomenclature consistently throughout the manuscript.
|
||
|
Comments 2: Population frequency comparisons require more sources and should specify whether data originate from genotyping or phenotyping. No statistical analyses are provided to evaluate the strength of the observed differences. Response 2: Thank you for pointing this out. We agree with the reviewer that frequency comparisons require more sources. We have now added Geoff Daniel’s textbook Human Blood Groups, 3rd edition, 2013, as additional resource (Table 1). We also agree with the reviewer that specifying whether data originate from genotyping or phenotyping is essential. However, population frequencies we have referred to (FactsBook and Geoff Daniels’s Human Blood Groups textbook), do not mention explicitly for each antigen whether the population frequencies originate from genotyping or phenotyping but the following statement in the second chapter (Organization of the data, page 18) of FactsBook can be interpreted as the antigen frequencies originate from phenotyping: “Phenotypes The prevalence of phenotypes associated with the blood group system, the null phenotype, and any unusual phenotypes (unusual in expression, not in prevalence) are given. In general, the figures given are for Caucasian populations (northern European), because that is the best studied group by hemagglutination but where possible and where particularly relevant due to differences between populations, other ethnic groups have also been included. Information was obtained from original publications, Blood Groups in Man9, and the AABB Technical Manual5. The numbers are an average estimate.” Reference 9: Race, R.R., Sanger, R., 1975. Blood Groups in Man, sixth ed. Blackwell Scientific Publications, Oxford, UK. Reference 5: Roback, J.D. (Ed.), 2011. Technical Manual, seventeenth ed. American Association of Blood Banks, Bethesda, MD. We now mention this in Table legend of Table 1 on page 7, lines 230-235. We have also run statistics to evaluate the strength of the observed differences of population frequency comparisons: revised Table 1
|
||
|
Comments 3: Lack of access to ethnic background data limits interpretation. A probabilistic analysis based on Rh haplotypes could have been performed.
Response 3: We thank the reviewer for this valuable suggestion. In the revised manuscript, we have now included a full probabilistic analysis of Rh haplotypes, integrating the observed distribution of combined RhD serology and RHCE genotyping for all 2 501 donors. We have now 2508 blood donors in the study. After excluding seven “no calls”, remaining 2501 donors´ data are used in this analysis. Haplotype frequencies (R¹, R², R⁰, Rᶻ, r, r′, r″) were estimated under Hardy–Weinberg equilibrium using the Excel Solver (GRG Nonlinear algorithm) to minimize the sum of squared errors (SSE) between observed and expected CDE phenotype proportions. The resulting frequencies (Supplementary Table S2) were consistent with those reported for Northern European populations, confirming that our donor cohort represents a genetically coherent population despite a higher proportion of RhD-negative donors.
Comments 4: Management of no calls: samples were not retested for economic reasons, which may bias population-level distributions. Why not exclude the donor’s entire genotype when a no call occurred for one antigen?
Response 4: Thank you for pointing this out. We think excluding a donor´s entire genotype when a “no call” occurred for one or two antigen(s) also may bias population-level distributions regarding the remaining 35-36 antigens without “no calls”. We hope therefore it will be acceptable for the reviewer that we can keep the original setup. When we calculated antigen frequencies, the "no calls" were excluded from the denominator. Serological results were used in routine but not when calculating antigen frequency analyses. This strategy is now explained under Materials and Methods: under RBC Genotyping on page 4, lines 176-179 and under Statistical Analysis on Page 5, lines 208-209.
Comments 5: These technical issues highlight the importance of complementary serological testing, especially when only one genotyping method is used without sequencing confirmation.
Response 5: Thank you for pointing this out. We agree with this comment. It would be the preferred strategy to perform sequencing to investigate discrepancies. Serology to start with and as a complementary test is essential and has been performed for each discrepancy case except for one as shown in Table 7. To solve discrepancies, we also need to be more proactive in using the other genotyping platform (RBC-FluoGene vERYfy and Rare kits (inno-train)) available at our laboratory. We use RBC-FluoGene kits mainly for RBC genotyping of patients in routine due to practical reasons. ID CORE XT is run approximately once a month, while we prioritize testing patient samples within a week when a patient requires transfusions. We then use RBC-FluoGene with smaller number of tests in kits. We have now discussed these issues in Discussion (page 14, lines 485-489),
Comments 6: Variants detected in 91 donors were not confirmed with a second method (e.g., sequencing, another commercial kit, Fluogene, or in-house PCR).
Response 6: Thank you for pointing this out. We do not have an in-house PCR method for detection of variants. We use confirmatory tests like sequencing and/or another commercial kit during validation processes, in addition to discrepancy investigations. Once a test is validated, we do not confirm variants detected in blood donors using a second method, because we consider that the actual method is trustworthy. We had explained our strategy under Materials and Methods in our original manuscript (page 4, lines: 181-182) and mentioned this drawback/limitation under Discussion in our original manuscript, page 15, lines 498-500. On the other hand, sequencing requires shipping samples to a reference laboratory. Sequencing at Grifols was used only once in the study period in discrepancy investigation.
Comments 7: For variants (Table 5), an additional column would be useful to indicate transfusion impact and how such donors are handled in product selection.
Response 7: Thank you for this comment. We have now added a new column to indicate how donors with variants were handled in product selection. Including the transfusion impact of variants would also be useful but is beyond the scope of our study.
Comments 8: ABO genotyping was not performed (not a limitation). However, the absence of RhD genotyping is problematic given the existence of variants with clinically significant transfusion impact. A complementary technique (e.g., Fluogene or similar) would have been appropriate, and would in any case be necessary if this strategy is to be deployed.
Response 8: Thank you for pointing this out. We agree with this comment. We may for example miss DEL donors. However, economic concerns and personal resources are the limiting factors. RhD genotyping of 2508 blood donors using FluoGene would have costed a lot. On the other hand, realizing that we actually have blood donors with other ethnic origins, we should consider preparing a strategy to identify and RhD genotype especially those donors. Since this study focuses on blood donors, we have not mentioned that we perform RhD genotyping routinely, when serological typing gives a £2+ result yet only in pregnant women, females of childbearing potential, considering that they may later become pregnant and/or patient needing transfusion. We then use D weak or CDE eXtend kits (FluoGene). A donor will be considered RhD positive regardless of how weak the serological typing is. Blood units from such a donor will not be transfused to an RhD negative patient. Weak RhD types 1, 2 and 3 are considered RhD positive since these patients and pregnant women are not at risk of becoming D-alloimmunized in pregnancy and due to RhD positive transfusions. Individuals with other RhD variants are considered RhD negative and will be ensured RhD negative blood components and receive anti-D immunoglobulin when non-invasive fetal RHD genotyping determines an RhD positive fetus. We have now mentioned and discussed the shortcomings of the absence of RhD genotyping in Discussion (page 14-15, lines 491-498, and page 12, lines: 346-351). And also in Conclusion (page 16, lines 518-521).
Minor limitations: Comment 9: Ensure consistency across tables. Response 9: Thank you for pointing this out. We have now uniformed the tables.
Comment 10: The organizational and economic impact remains insufficiently addressed; such a study would be highly relevant for drawing broader conclusions in this area. Response 10: Thank you for pointing this out. We agree with this comment. We have now added these aspects under Discussion, on page 11, lines 308-316.
|
||
|
Point 1: (x) The English is fine and does not require any improvement.
|
||
|
Response to point 1: Thank you.
|
||
Reviewer 2 Report
Comments and Suggestions for Authors
"In line 89 of the introduction, the authors state: 'Blood donors are also routinely phenotyped for C, c, E, e and K. Mixed field reactions due to recent blood transfusion may be difficult to interpret.' However, as donors who have recently received transfusions are typically deferred from donating, this statement may be misleading. While the intention is clear, the authors should consider rephrasing it to better reflect current donor eligibility criteria."
The selection of antigens in the technique—such as C, c, E, e, and K—reflects a balance between clinical relevance, prevalence, and technical feasibility. These antigens are among the most commonly implicated in transfusion reactions and alloimmunization, making them high-priority targets for routine phenotyping.
However, the exclusion of other clinically significant antigens such as Vel, Jra, and others raises important considerations:
- Clinical Impact: Antigens like Vel and Jra, although less common, can be highly immunogenic. Their absence from routine screening may pose risks in specific patient populations, especially those with rare blood types or multiple alloantibodies.
- Technical Limitations: Some antigens may be excluded due to limitations in assay design, availability of reagents, or the complexity of detecting low-frequency variants. Including them could increase the cost and reduce throughput.
- Population-Specific Relevance.
- Cost vs. Benefit: Including a broader range of antigens increases the comprehensiveness of the test but also raises costs and complexity. The current selection likely reflects a compromise aimed at maximizing clinical utility while maintaining efficiency.
In the Materials and Methods section (line 163), the authors define “no call (NC)” and “unknown” results but do not specify how these cases were handled during the study. However, in the Discussion section (line 257), they mention that initial 'no calls' (0.14%) were traced back to the use of an incorrect thermal block, which affected PCR performance. After correcting the issue, the problem was resolved, but the affected samples were not reanalyzed due to cost constraints; instead, serologic results were used. This important methodological decision should be clearly stated in the Materials and Methods section to ensure transparency and reproducibility."
In the Materials and Methods section (line 172), the authors describe their protocol for resolving discrepancies, including the possibility of sending samples to Grifols for further analysis. However, the Results section does not specify how many samples were actually sent, nor does it mention the turnaround time for receiving results. This information is important for assessing the practicality and efficiency of the workflow. Additionally, the Discussion section could benefit from addressing the potential drawbacks—or logistical challenges—of outsourcing sample analysis, such as delays, costs, or dependency on external facilities. Including these details would enhance transparency and help readers better understand the operational implications of the technique.
In the Results section (line 235), the authors state: “Additionally, 12 discrepancies were related to Fya (data not shown). In two other samples there was…” However, it is unclear whether the subsequent lines refer to the same 12 Fya-related discrepancies or to separate cases. For clarity, the authors should explicitly indicate whether these two samples are part of the 12 previously mentioned or represent additional discrepancies. This would help readers better understand the scope and nature of the findings.
"In the discussion and conclusion sections, the authors should critically examine and emphasize the limitations of the technique, including its restricted applicability to certain antigen types (e.g., exclusion of ABO and RhD), constraints in throughput, and associated costs."
Author Response
REVIEWER 2:
Thank you very much for taking the time to review this manuscript. Please find the detailed responses below and the corresponding revisions/corrections in track changes in the re-submitted files.
|
Comment 1: "In line 89 of the introduction, the authors state: 'Blood donors are also routinely phenotyped for C, c, E, e and K. Mixed field reactions due to recent blood transfusion may be difficult to interpret.' However, as donors who have recently received transfusions are typically deferred from donating, this statement may be misleading. While the intention is clear, the authors should consider rephrasing it to better reflect current donor eligibility criteria."
|
|
Response 1: Thank you for pointing this out. We agree with this comment. In lines from 86 of the Introduction, we mention both blood donors and patients. The intention of mentioning mixed field reactions due to recent blood transfusions was to describe this phenomenon in patients but we agree that the statement can be misleading/misunderstood, therefore, we have rephrased the statement (page 2, line 92 and page 3, line 99). |
Comment 2: The selection of antigens in the technique—such as C, c, E, e, and K—reflects a balance between clinical relevance, prevalence, and technical feasibility. These antigens are among the most commonly implicated in transfusion reactions and alloimmunization, making them high-priority targets for routine phenotyping.
However, the exclusion of other clinically significant antigens such as Vel, Jra, and others raises important considerations:
- Clinical Impact: Antigens like Vel and Jra, although less common, can be highly immunogenic. Their absence from routine screening may pose risks in specific patient populations, especially those with rare blood types or multiple alloantibodies.
- Technical Limitations: Some antigens may be excluded due to limitations in assay design, availability of reagents, or the complexity of detecting low-frequency variants. Including them could increase the cost and reduce throughput.
- Population-Specific Relevance.
- Cost vs. Benefit: Including a broader range of antigens increases the comprehensiveness of the test but also raises costs and complexity. The current selection likely reflects a compromise aimed at maximizing clinical utility while maintaining efficiency.
Response 2: Thank you for pointing out all these issues. Our response below is meant to cover the reviewer´s comments regarding clinical impact, technical limitations, population-specific relevance and cost vs. benefit (points 1-4):
As the reviewer states, (which we totally agree) antigens not covered by a standard kit, although these antigens are less common, may pose risks for the patient. As we have a couple of patients with anti-Vel and considering that Vel negativity is more common in the Nordic countries, we, at an earlier phase, discussed Vel screening of our blood donors using a dedicated FluoGene kit (FluoGene Vel-Screen) but concluded that it would be too costly. The Rare kit (FluoGene) has
Jra. A newer FluoGene vERYfy eXtend kit, which is a combination of vERYfy and Rare kits plus some other antigens, including Vel, is also available, but using these kits as standard for donors will again be costly. Besides, FluoGene is not a high-throughput platform. Patients with several antibodies are probably at less risk, as their PRBCC units will only be issued following a negative cross-match letting us to capture for example anti-Vel and probably other clinically significant antibodies unless the antibody level is under the detection level. Otherwise (if the antibody level is under the detection level), the patient may risk a delayed hemolytic transfusion reaction. We have now discussed these challenges on page 12, lines: 346-349, and on page 16, line: 519.
Comment 3: In the Materials and Methods section (line 163), the authors define “no call (NC)” and “unknown” results but do not specify how these cases were handled during the study. However, in the Discussion section (line 257), they mention that initial 'no calls' (0.14%) were traced back to the use of an incorrect thermal block, which affected PCR performance. After correcting the issue, the problem was resolved, but the affected samples were not reanalyzed due to cost constraints; instead, serologic results were used. This important methodological decision should be clearly stated in the Materials and Methods section to ensure transparency and reproducibility."
Response 3: Thank you for pointing out these issues. In the Materials and Methods section, we have now added how samples with “unknown” results and “no calls” were handled during the study (page 4, line 176-179). These samples were not reanalyzed, and serological results, when available, were used instead not in this study setting, but in routine.
Comment 4: In the Materials and Methods section (line 172), the authors describe their protocol for resolving discrepancies, including the possibility of sending samples to Grifols for further analysis. However, the Results section does not specify how many samples were actually sent, nor does it mention the turnaround time for receiving results. This information is important for assessing the practicality and efficiency of the workflow. Additionally, the Discussion section could benefit from addressing the potential drawbacks—or logistical challenges—of outsourcing sample analysis, such as delays, costs, or dependency on external facilities. Including these details would enhance transparency and help readers better understand the operational implications of the technique.
Response 4: Thank you for pointing this out. We agree with this comment. As mentioned in the Materials and Methods section, shipping the sample has been the last option when a discrepancy could not be resolved by repeat serology and/or genotyping using RBC-FluoGene vERYfy and Rare kits. Sample 7 in Table 7 is the only sample sent to Grifols in the study period. Turnaround time was approximately one month from the sample was sent till the result was reported back to us. However, since Grifols requires fresh blood samples, we waited until the donor´s next donation. Until elucidation of the result, the blood units from the donor were treated as antigen-positive (C positive in this case, sample 7) not to risk alloimmunization of antigen-negative (C negative) patients. Our strategy is to consider a blood donor antigen-positive (not to risk immunizing an antigen-negative patient) in any case if there is doubt about the conclusion. We have now added specifications the reviewer has recommended in the Result section, page 9, lines 282-283. We have also mentioned the logistical challenges of outsourcing sample analysis including turnaround time among the other drawbacks in the Discussion section, page 15, lines 505-511.
Comment 5: In the Results section (line 235), the authors state: “Additionally, 12 discrepancies were related to Fya (data not shown). In two other samples there was…” However, it is unclear whether the subsequent lines refer to the same 12 Fya-related discrepancies or to separate cases. For clarity, the authors should explicitly indicate whether these two samples are part of the 12 previously mentioned or represent additional discrepancies. This would help readers better understand the scope and nature of the findings.
Response 5: Thank you for pointing this out. The two other samples were two separate samples/cases. These two samples are not part of the 12 previously mentioned discrepancies but represent additional discrepancies. We hope we now have clarified this in the Table legend, page 10, lines 286-287.
Comment 6: "In the discussion and conclusion sections, the authors should critically examine and emphasize the limitations of the technique, including its restricted applicability to certain antigen types (e.g., exclusion of ABO and RhD), constraints in throughput, and associated costs."
Response 6: Thank you for pointing this out. We have now critically discussed and emphasized the limitations of the technique, including its restricted applicability to certain antigens like ABO and RhD in both the Discussion (page 12, lines 346-350 and page 14-15, lines 491-498) and Conclusion: page 16, lines 518-523.
Round 2
Reviewer 2 Report
Comments and Suggestions for Authors
Well done, congratulations